# HPO-RL-Bench: A Zero-Cost Benchmark for HPO in Reinforcement Learning

**Gresa Shala**[1]  **Sebastian Pineda Arango**[1]  **André Biedenkapp**[1]  **Frank Hutter**[3, 1]
**Josif Grabocka**[2]

[1]University of Freiburg
[2]University of Technology Nuremberg
[3]ELLIS Institute Tübingen

**Abstract**  Despite the undeniable importance of optimizing the hyperparameters of RL algorithms, existing state-of-the-art Hyperparameter Optimization (HPO) techniques are not frequently utilized by RL researchers. To catalyze HPO research in RL, we present a new large-scale benchmark that includes pre-computed reward curve evaluations of hyperparameter configurations for six established RL algorithms (PPO, DDPG, A2C, SAC, TD3, DQN) on 22 environments (Atari, Mujoco, Control), repeated for multiple seeds. We exhaustively computed the reward curves of all possible combinations of hyperparameters for the considered hyperparameter spaces for each RL algorithm in each environment. As a result, our benchmark permits zero-cost experiments for deploying and comparing new HPO methods. In addition, the benchmark offers a set of integrated HPO methods, enabling plug-and-play tuning of the hyperparameters of new RL algorithms, while pre-computed evaluations allow a zero-cost comparison of a new RL algorithm against the tuned RL baselines in our benchmark.

## 1 Introduction

Reinforcement Learning (RL) applications have made headlines in the past decade, with breakthroughs in a variety of domains such as game playing [see, e.g., 41, 55, 9, 44], robotics [3] or real-world tasks [6, 16]. These demonstrations of the capabilities of RL algorithms have fuelled a surge of interest in the research community. In spite of achieving impressive results, RL remains highly sensitive to hyperparameter configurations and implementation details [31, 21, 4, 29]. Bundled together with a typically high cost of hyperparameter optimization (HPO) experiments, this makes manual tuning of RL agents highly error-prone, tedious and requires heaps of expert knowledge.

To catalyze research in the field of HPO for RL we introduce **HPO-RL-Bench**, the first zero-cost HPO benchmark which contains pre-computed reward curves for six popular model-free RL algorithms across 22 environments (illustrated in Figure 1). We consider three distinct classes of environments from OpenAI Gym [12], Atari [7], Classic Control, and MuJoCo [57] and focus on six popular RL algorithms: PPO [53], DDPG [36], A2C [40], SAC [27], TD3 [25], and DQN [41]. We evaluate hundreds of distinct hyperparameter configurations for each RL algorithm and each environment, repeated for 10 seeds. Overall, our benchmark incorporates ca. $200K$ training runs. Furthermore, the benchmark offers a set of 7 HPO techniques, which have been evaluated in all environments for all methods.

We believe the benchmark provides unique properties for HPO researchers but also for the RL community:

- An HPO researcher can evaluate a novel HPO technique on 6 hyperparameter spaces and 22 environments, and compare it to 7 HPO baselines, with zero costs.

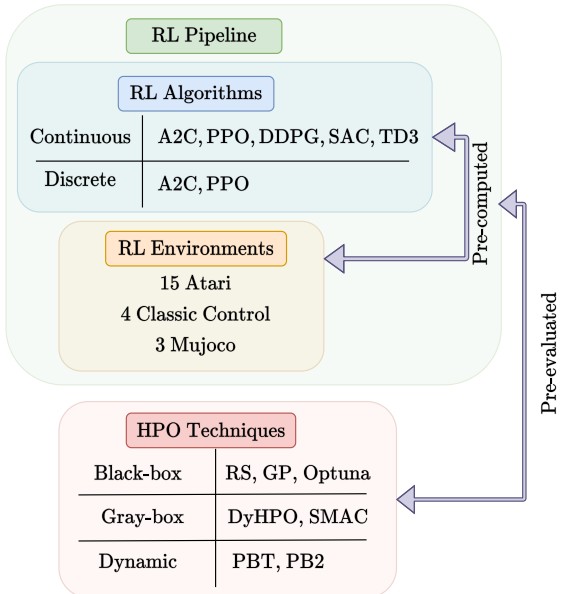

Figure 1: HPO-RL-Bench offers exhaustively pre-computed reward curves for hyperparameters of RL algorithms, as well as pre-evaluated results of HPO techniques.

- An RL researcher working on a new algorithm can compare against 6 baseline RL algorithms in 22 environments, with zero costs. Additionally, they can tune the hyper-parameters of the new RL algorithm with a few lines of code using the 7 provided HPO methods.

Furthermore, as dynamic adaptation plays an important role in deep RL [47, 42], for a subset of algorithms and environments, we evaluate 729 schedules of hyperparameters with distinct switching points on a subset of considered environments, which allows us to give a head-to-head comparison between online and gray-box HPO methods.

To the best of our knowledge, we are the first to provide a zero-cost HPO benchmark for model-free RL algorithms.

## 2 Related Work

**Reproducibility of RL**. Comparison and reproducibility of RL experiments remain difficult. The sensitivity of RL algorithms to hyperparameters [31, 28] is exacerbated by implementation details that can strongly influence their performance [21, 4, 29]. Further, the cost of typical RL experiments causes studies to often only compare performances on a handful of trials, which is most often not sufficient for a clear comparison [1]. Our pre-computed learning curves provide a set of baselines against which RL practitioners can easily and fairly compare their RL algorithms but also study HPO methods.

**HPO and Zero-Cost Benchmarks**. Tabular benchmarks have been proposed for other important problems in the (Auto)ML community where evaluations are expensive. For example, in the field of neural architecture search [NAS; 20], evaluating the performance of deep learning architectures can quickly become very resource intensive. Thus, to make novel NAS methods easily and cheaply comparable and reproducible, tabular benchmarks [60, 61, 17, 54, 39] have become an important tool for NAS research [38] and allowed rapid development of NAS methods. Similarly, tabular benchmarks play important roles in the fields of HPO [11] and gray-box optimization [34, 33], see, e.g., HPO-B [48] and HPOBench [18]. While RL is similarly or potentially even more expensive to

train and evaluate, to the best of our knowledge, any tabular benchmarks on which HPO for RL could be studied have not yet been proposed.

**HPO for RL**. While there are various parts of the RL pipeline that could be automated, e.g., the choice of algorithm or environment components, in this work we focus on hyperparameter optimization (HPO) for RL.[1] One of the best-understood and studied hyperparameters of RL is the discounting factor $\gamma$. For example, it is known that smaller values of $\gamma$ lead to faster convergence but might result in myopic policies [10]. Increasing the discounting value over time can drastically speed up learning [23]. François-Lavet et al. [23] showed that simultaneously decreasing the learning rate while increasing $\gamma$ can improve learning speeds even further. Still, for most algorithms and their hyperparameters, it is not clear or understood whether they are best adapted during training or whether they should stay fixed [47], and how they influence the learning dynamics in general. A recent work however by Mohan et al. [42] however demonstrated that the hyperparameter landscapes of RL agents changes over time, giving evidence that dynamic tuning is likely needed for RL. Thus, it is common for RL practitioners to use some default configuration without exploring different types of HPO methods. To alleviate users from having to manually tune their RL agent, various HPO methods for RL have been proposed [see, e.g., 32, 14, 52, 46, 43, 24, 5]. Further, Eimer et al. [19] showed that RL hyperparameter landscapes appear smooth, thus automated HPO methods are capable of producing better-performing RL agents than hyperparameter sweeps or grid searches. Still, HPO methods have not yet found widespread adoption by the RL community.

**Benchmarks for RL**. There exists a plethora of benchmarks to evaluate RL algorithms [see, e.g., 57, 7, 12, 15, 51, 45]. However, these are designed with RL in mind, not HPO for RL. Such benchmarks provide environments for the agents to interact with and collect training examples. This makes them prohibitive for use in HPO experiments as any RL agent being optimized will still have to compute expensive training updates. Our proposed benchmark differs from RL benchmarks in providing precomputed reward curves of already trained agents for specific hyperparameter configurations. As a result, both RL and HPO researchers can benefit from HPO-RL-Bench.

## 3 Benchmark Description

The benchmark comprises recorded episodic reward-curves for five commonly used RL algorithms PPO [53], A2C [40], DDPG [36], SAC [27], and TD3 [25] on 22 environments (see Figure 7 in Appendix A). For each algorithm, we consider the static configuration space listed in Table 1. Furthermore, for the Classic Control and MuJoCo environments, we consider extended versions of the static configuration search spaces of PPO and A2C, by adding architectural hyperparameters. We list the extended configuration spaces in Table 2 of Appendix C.1. For PPO, SAC, and TD3, we additionally consider a dynamic search space in which hyperparameters can change at discrete time-steps *while* the agent is training (highlighted in blue text in Table 1). Our HPO-RL-Bench contains the recorded evaluation episodic reward-curves for all training runs of each agent with all possible combinations of hyperparameters in the chosen configuration spaces.

The considered OpenAI Gym [12] environments consist of 15 Atari [7] games, 4 Classic Control problems, and 3 MuJoCo [57] tasks. Most environments have a discrete action space. Only the Classic Control task *Pendulum* and the MuJoCo environments *Ant*, *Hopper* and *Humanoid* have continuous action spaces. All 15 Atari games have image-based state representations, whereas the Classic Control and MuJoCo environments have vector-based state representations. Most environments have dense reward signals, only the games *Bowling*, *Enduro*, *Pong*, *Skiing* and *Tennis* have mostly sparse reward signals. We computed episodic reward curves with static configurations on all environments. For the dynamic configuration schedules, we used only the Classic Control tasks and the Enduro game.

---

[1]For a comprehensive survey on HPO in RL we refer to Parker-Holder et al. [47].

Table 1: Configuration spaces of the considered RL algorithms. **Blue bold faced** entries show the subset considered for the dynamic variant.

| Algo. | Hyperaram. | Values |
|---|---|---|
| PPO | lr ($\log_{10}$) | $-6, -5, -4, -3, -2, -1$ |
|  | $\gamma$ | $0.8, 0.9, \mathbf{0.95, 0.98, 0.99,} 1.0$ |
|  | clip | $0.1, 0.2, 0.3$ |
| A2C | lr ($\log_{10}$) | $-6, -5, -4, -3, -2, -1$ |
|  | $\gamma$ | $0.8, 0.9, 0.95, 0.98, 0.99, 1.0$ |
| DQN | lr ($\log_{10}$) | $-6, -5, -4, -3, -2, -1$ |
|  | $\gamma$ | $0.8, 0.9, 0.95, 0.98, 0.99, 1.0$ |
|  | $\epsilon$ | $0.1, 0.2, 0.3$ |
| DDPG | lr ($\log_{10}$) | $-6, -5, -4, -3, -2, -1$ |
|  | $\gamma$ | $0.8, 0.9, 0.95, 0.98, 0.99, 1.0$ |
|  | $\tau$ | $0.001, 0.005, 0.01$ |
|  | n_layers | $1, 2, 3$ |
|  | n_units | $32, 64, 128, 256$ |
| SAC | lr ($\log_{10}$) | $-6, -5, -4, -3, -2, -1$ |
|  | $\gamma$ | $0.8, 0.9, \mathbf{0.95, 0.98, 0.99,} 1.0$ |
|  | $\tau$ | $0.001, 0.005, 0.01$ |
|  | n_layers | $1, 2, 3$ |
|  | n_units | $32, 64, 128, 256$ |
| TD3 | lr ($\log_{10}$) | $-6, -5, -4, -3, -2, -1$ |
|  | $\gamma$ | $0.8, 0.9, \mathbf{0.95, 0.98, 0.99,} 1.0$ |
|  | $\tau$ | $0.001, 0.005, 0.01$ |
|  | n_layers | $1, 2, 3$ |
|  | n_units | $32, 64, 128, 256$ |

### 3.1 Data Collection

For the RL algorithms, we used the implementations from **stable-baselines3** [50]. To be able to provide uncertainty estimates, we ran each configuration (schedule) for ten seeds on a compute cluster using RTX 2080 GPUs. We trained all agents for $10^6$ steps on each environment and evaluated the performance for 10 episodes every $10^4$ steps. The total cost of creating the benchmark amounts to 274 320 GPU hours, or 31.3 GPU years of computational resources. We believe that, as is the case with tabular NAS benchmarks, this compute cost will be more than amortized by the benchmark's many potential uses.

When training PPO, SAC, and TD3 with configuration schedules, to avoid a combinatorial explosion,[2] we limited the configuration space to two hyperparameters with three values each and used two discrete switching points after $3 \cdot 10^5$ and $6 \cdot 10^5$ training steps elapsed. This gives rise to a configuration space of $(3^2)^3 = 729$ distinct configuration schedules, all of which were evaluated on five environments for five seeds each. Further details are provided in Appendix B.

---

[2]The original PPO configuration space with a single switch would already have required $(6 \cdot 6 \cdot 3)^2 = 11\,664$ evaluations and with two switching points $(6 \cdot 6 \cdot 3)^3 = 1\,259\,712$ evaluations. Similarly, in the pruned space, a third switch would already result in $(3^2)^4 = 6\,561$ schedules.

## 3.2 API for HPO-RL-Bench 1.0

To ease the benchmark's accessibility, we provide an API which is freely accessible at `https://github.com/releaunifreiburg/HPO-RL-Bench`. Once the data is downloaded, a few lines of code suffice to query the metrics of a hyperparameter configuration for a given environment-search space combination. An example of doing this when optimizing with random search is given in Listing 1 in the Appendix. It is possible to query any hyperparameter configuration for all the listed algorithms in Table 1 and environments in Figure 7. Also, the user can query the dynamic or static spaces by modifying the respective attribute in the benchmark object (benchmark.static = True/False). When querying a dynamic configuration, the user must provide the list of hyperparameter values that are used in the schedule, where switches are possible at 300k and 600k training steps (see Listing 2 in the Appendix). In our GitHub repository, we provide further examples on advanced ways to query the API to avoid creating an object for every environment/search-space combination. Moreover, we provide examples of how to couple its functionality with HPO optimizers.

## 4 Experiments

In this section, we demonstrate the usefulness of our benchmark by addressing understudied problems in the (Auto)RL community. We begin by studying the hyperparameter importance in the covered configuration spaces before providing a comprehensive comparison of existing HPO methods in optimizing the hyperparameters of six popular RL algorithms. Finally, we end this section by evaluating the competitiveness of our chosen configuration spaces.

### 4.1 Hyperparameter Importance for RL Agents

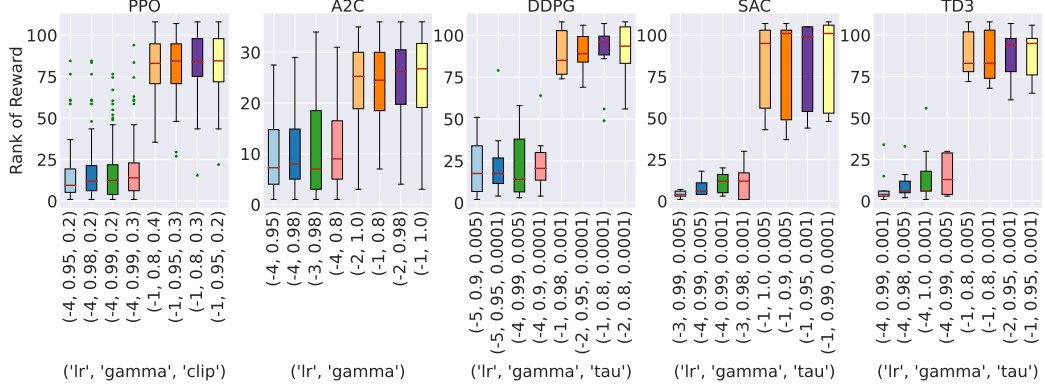

Figure 2: Average rank of the final reward across environments (four best and worst configurations). The illustrations depict: red line=median, green dots=outliers, colored box=inter-quantile range(IQR) from the first quantile(Q1) to the third quantile(Q3) of the data, whiskers=extended IQR by 1.5x.

To determine the importance of hyperparameters for the considered RL algorithms, we address the following questions. i) Which static hyperparameter configurations result in the best final episodic evaluation reward for each considered algorithm? ii) Which hyperparameters are more important, i.e. have a higher influence on the final episodic evaluation reward?

To answer the first question we compute the average rank based on the final evaluation return of all considered static configurations per environment and for each configuration space separately. Based on these averages, we select the four best and worst configurations in each configuration space. Figure 2 depicts the results as a box plot, where lower ranks indicate better final rewards. We present the results for the DQN search space in Figure 18a in the Appendix. Generally, on average, lower learning rates result in better final rewards than configurations with high learning rates.

Further, our results indicate that PPO and A2C configurations are less robust than those of SAC, TD3, DDPG and DQN, as generally poorly-performing configurations can in some environments result in very good final rewards and vice versa. Contrary to practitioners' hopes, there exists no *silver bullet* hyperparameter configuration that is optimal in the vast majority of the environments. The take-home message is that optimal hyperparameters are environment-specific and must be carefully tuned.

To answer the second question we make use of the fANOVA hyperparameter importance method [30], which aims to quantify how strongly the change in a hyperparameter value influences the observed final episodic evaluation reward. fANOVA attributes higher importance to those hyperparameters that have a stronger influence on the final return. In Figure 3 we compute the average rank over all environments. We present the results for the DQN search space in Figure 18b in the Appendix. Our results confirm that the learning rate is instrumental in achieving optimal performance for the considered algorithms. Furthermore, $\gamma$ is very influential in the case of SAC, TD3 and DQN. We include a more detailed analysis of the full configuration spaces including the architectural hyperparameters in Appendix C.2.

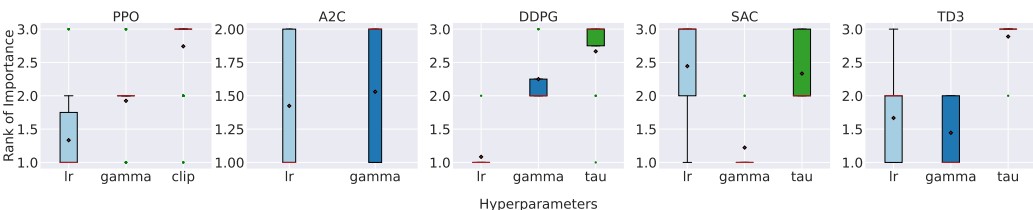

Figure 3: Hyperparameter importance per search space (red diamond=mean, red line=median, green dots=outliers, colored box=inter-quantile range(IQR) from the first quantile(Q1) to the third quantile(Q3) of the data, whiskers=extended IQR by 1.5x.).

## 4.2 HPO for RL

To demonstrate how HPO methods for RL or other AutoRL approaches could leverage our novel benchmark we provide a comprehensive comparison of existing HPO methods and evaluate their usefulness for RL. To this end we evaluate the following baselines:

**Random Search** (RS) is a simple and standard HPO baseline. It selects hyperparameter configurations uniformly at random in the given search space.

**Bayesian optimization with Gaussian Proccesses** [GP; 56] is another standard HPO baseline. This baseline uses GPs as the surrogate model in standard black-box Bayesian optimization. We used the implementation in GPytorch [26] with a Matern 5/2 kernel.

**SMAC4MF** [SMAC; 37] implements a variant of the gray-box optimizer BOHB [22] which combines Hyperband [35] with Bayesian optimization [BO; 56]. The Hyperband component allows to quickly discard under-performing configurations on smaller budgets (i.e., few epochs or number of training samples), whereas the BO component identifies well-performing regions of hyperparameters from which to sample. SMAC4MF differs from the original BOHB by fitting a Random Forest for the BO component.

**DyHPO** [59] is a gray-box method that uses a deep kernel [58] with a convolutional neural network that embeds the reward curves and incorporates budget information in its acquisition function. This allows DyHPO to dynamically decide with which budget the next configuration should be evaluated.

**Optuna** [2] is a popular hyperparameter optimization framework. Following Raffin [49], who used it to tune hyperparameters of stable-baselines3, we used Optuna with TPE [8] as a surrogate.

**Population Based Training** [PBT; 32] is an evolutionary method for HPO that allows to dynamically change hyperparameters during training. PBT maintains a population of RL agents.

Every $N$ steps (a user-defined value) the worst members in the population are replaced with the best ones. Simultaneously the hyperparameters of these replaced agents are perturbed to explore if new hyperparameter values might improve the performance further. To work well, PBT typically requires large populations between 40 and 80 members.

**Population based bandits** [PB2; 46] extends the PBT framework and replaces the random hyperparameter perturbations with predictions from a time-varying Gaussian process. This change enables PB2 to perform a more informed search over hyperparameters. As a consequence, PB2 typically requires drastically fewer members (i.e., only 4 to 8) in the population compared to its predecessor PBT.

**Setup.** On the static benchmark, for the PPO and A2C search spaces defined in Table 1, we evaluated RS, GP, Optuna, SMAC, and DyHPO for a budget of 10 full training runs for each algorithm-environment-seed triple (i.e. $10^7$ training steps). For the DDPG, TD3, and SAC static search spaces, as well as the extended versions of the PPO and A2C search spaces we evaluated the aforementioned baselines for a budget of 50 full training runs for each algorithm-environment-seed triple. Initially, RS, GP, Optuna, SMAC, and DyHPO start the search with the same 4 hyperparameter configurations sampled uniformly at random. As PBT and PB2 are designed to optimize hyperparameters dynamically, we evaluate them on the dynamic version of our benchmark. We denote these results with the labels D-PBT, and D-PB2. Additionally, we evaluated PBT and PB2 by actually training the suggested configurations using the same pipeline as the one we used for HPO-RL-Bench. We used the same search space as the one indicated in Table 1. We ran all versions of the population-based baselines (i.e. PBT, PB2, D-PBT, D-PB2) with a population size of 8. For all baselines, we report the average rank of the evaluated methods across environments (lower is better). We group environments into Atari, Mujoco, and Control following the description of Section 3, and present group-aggregated results. Additional results on a per-environment basis are included in Appendix C.

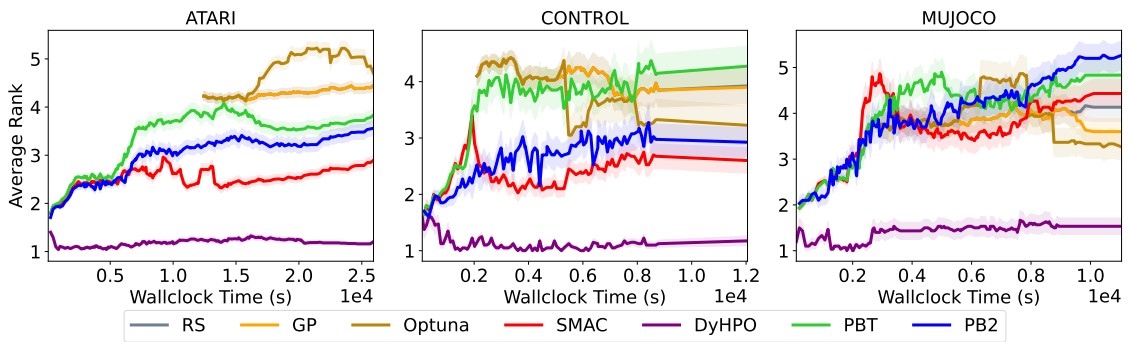

Figure 4: Average Ranks of the performance of the baselines for the PPO configuration space.

**Results.** In all configuration spaces, we observe that the online HPO methods PBT and PB2 perform well in the beginning, but given enough time, RS, GP, Optuna, SMAC, and DyHPO find static hyperparameter configurations that outperform their schedules (see Figures 4 and 5a). Generally, PBT and PB2 perform similarly to each other. On the smaller configuration space for A2C (see Figure 8 in Appendix C) PBT outperforms PB2 on average in the MuJoCo and Classic Control environments, whereas PB2 outperforms PBT on Atari environments. For the PPO configuration spaces, PB2 clearly outperforms PBT in the Classic Control environments. In conclusion, PBT and PB2 are efficient in terms of discovering configurations under limited budgets, but on the A2C search space (see Figure 8 in Appendix C) even naive Random Search outperforms them when more than about 3 hours of HPO time per environment is available. Such findings indicate that the

community needs novel HPO methods that both converge quickly given a low HPO budget, but also remain competitive when more computing time is available.

Moreover, the results indicate that SMAC and DyHPO, the gray-box HPO baselines in our collection, are amongst the best-performing methods across time steps.

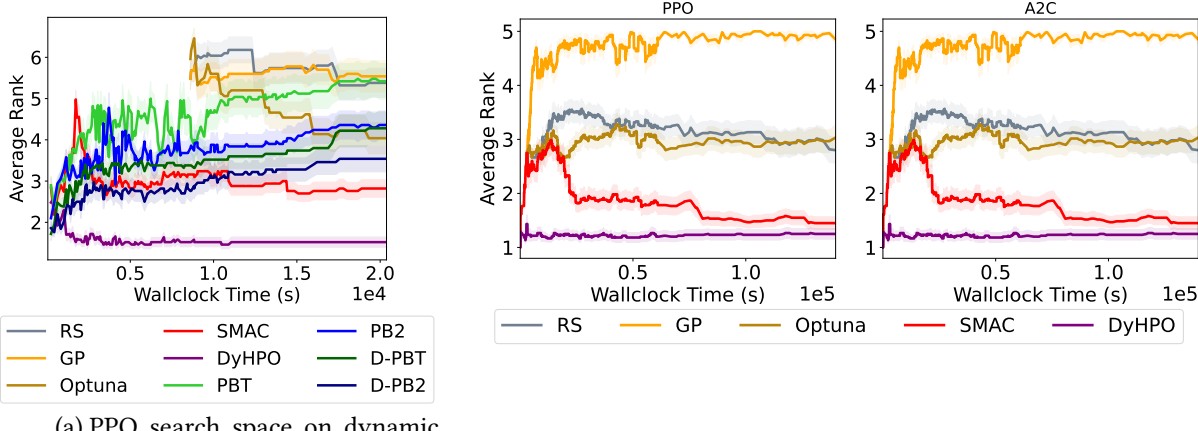

(a) PPO search space on dynamic benchmark.

(b) Extended search spaces of PPO and A2C on MuJoCo.

Figure 5: Average Ranks of the performance of the baselines for: 5a optimizing PPO on the dynamic benchmark, and 5b the extended versions of the PPO and A2C search spaces on the MuJoCo environments.

Additionally, we show the performance of the baselines on extended versions of the PPO and A2C static search spaces for the MuJoCo environments in Figure 5b. Table 2 in Appendix C.1 includes a detailed description of these search spaces. As it is not trivial for PBT and PB2 to optimize architectural hyperparameters, we only compare RS, GP, Optuna, SMAC, and DyHPO. DyHPO and SMAC outperform the black-box optimization methods. We show the performance of the baselines on the DDPG, TD3, SAC and DQN search spaces in Figures 9, 10, and 11 as well as the extended versions of the PPO and A2C static search spaces for the Classic Control environments in Figure 13 in Appendix C.

### 4.3 Validating the Usefulness of the Considered Configuration Space

In the design of the configuration space for HPO-RL-Bench we put the focus on a small set of hyperparameters. To demonstrate that tuning the hyperparameters in this small space yields strong performance, we provide additional experiments for the static as well as the dynamic search spaces.

The authors of stable-baselines3 provide tuned hyperparameters in the popular RL-Zoo framework [49]. These tuned configurations are obtained by running Optuna [2] on a large configuration space containing 9 to 13 hyperparameters (out of which 2 to 3 are searched in a continuous range, depending on the RL algorithm).[3] For example, the RL-Zoo3 PPO configuration space considers a total of 12 hyperparameters, out of which 3 (including the learning rate) are searched in a continuous space. The considered hyperparameter values for the discounting factor $\gamma$ and the clip range largely overlap with our configuration space.

To showcase that our configuration space is competitive, we take the hyperparameter tuning setup of RL-Zoo and apply it to our chosen configuration space, i.e., we run Optuna on our configuration space. We compare the performance of the found hyperparameter configurations against those determined on the original RL-Zoo configuration space and our benchmark's oracle

---

[3]We refer to the RL-Zoo github for the full configuration spaces https://github.com/DLR-RM/rl-baselines3-zoo/blob/master/rl_zoo3/hyperparams_opt.py

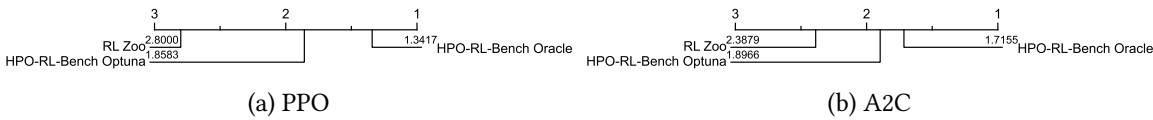

(a) PPO                  (b) A2C

Figure 6: Critical Difference diagram for final evaluation return of hyperparameter configurations suggested by RL-Zoo, those optimized by Optuna on the HPO-RL-Bench search space, as well as the best-performing hyperparameter configuration in HPO-RL-Bench.

(i.e., the best configuration for each environment). We create rankings of these comparisons for each of the RL algorithms and show the results in the critical difference diagrams of Figure 6 (for DDPG, SAC, TD3, and DQN see Figure 12 in Appendix C). The position along the x-axis presents the ranking of the final evaluation reward of the best-found configurations averaged over all environments and seeds. Thick horizontal lines indicate that there is no statistically significant difference between the rankings of the methods linked, according to the Wilcoxon-Rank test. It is clear that, while Optuna is capable of finding well-performing configurations in all considered configuration spaces, there is still room for improvement when comparing the performance to an oracle. Further, Optuna produced much better-performing configurations on our smaller configuration spaces than on the large RL-Zoo spaces (with a statistically significant difference for all search spaces). Interestingly, in the DDPG search space, tuning with Optuna on our configuration space resulted in highly performing configurations that could close the gap to the oracle such that the resulting ranking was not statistically significant. These results show that our chosen configuration space is meaningful and provides ample opportunity to study hyperparameter optimization methods on static search spaces.

Additionally, to get a better understanding of the dynamic configuration spaces and to facilitate a comparison between gray-box and online HPO methods on larger hyperparameter spaces, we compare to PBT and PB2 results for real (not precomputed) runs on the full configuration spaces. These results are denoted with the labels PBT and PB2, respectively whereas the D-PBT and D-PB2 results indicate the performance of PBT and PB2 on the subspace we evaluated exhaustively. Following Parker-Holder et al. [46], we used a population of 8 members for both PBT and PB2 for these experiments.

Figures 4 and 5a show that running PBT and PB2 on our benchmark yields very similar results to running the default PBT/PB2 implementations, therefore validating the correctness of the dynamic benchmark. Overall, our experimental results demonstrated that tuning the hyperparameters of RL algorithms is an open challenge and we believe this benchmark will be the *de facto* experimental protocol for innovating on more efficient HPO methods for RL, which would enable practitioners to deploy RL in an off-the-shelf manner on new environments.

## 5 Conclusion

We presented the first tabular HPO benchmark for RL. Our tabular benchmark drastically reduces the computational requirements for evaluating novel HPO methods in RL and, in turn, dramatically lowers the barrier to entry into this field of study. HPO-RL-Bench consists of evaluation episodic reward curves for six commonly used RL methods across a diverse set of 22 environments. In particular, counter to commonly provided tabular HPO benchmarks, our benchmarks allow studying configuration schedules through distinct switching points. We demonstrated the value of our benchmark to the HPO & RL communities by using it to evaluate commonly used HPO methods. Lastly, we showed how our benchmark can provide insights to RL practitioners about the influence of hyperparameters on an agent's performance and provides pre-computed baselines. We believe that our benchmark opens the door for the study of novel HPO methods and will help similarly advance the field as tabular benchmarks helped advance research in neural architecture search.

## 6 Broader Impact Statement

By providing pre-computed evaluations of hyperparameter configurations for six RL algorithms across diverse environments, HPO-RL-Bench addresses the underutilization of HPO techniques in RL research. This benchmark democratizes HPO research by enabling zero-cost experiments, fostering innovation, and accelerating progress in the field. Integration of various HPO methods streamlines model development and promotes transparency and reproducibility in RL research. However, there is a risk that HPO methods may become overfitted to our benchmark, compromising their generalization to markedly different environments. This limitation, inherent to any HPO benchmark, underscores that the results and insights derived from our benchmark should be interpreted as specific to the set of environments it encompasses.

**Acknowledgements**. This research was partially supported by TAILOR, a project funded by EU Horizon 2020 research and innovation programme under GA No 952215. André Biedenkapp and Frank Hutter acknowledge Robert Bosch GmbH for financial support. André Biedeankpp and Frank Hutter acknowledge funding by European Research Council (ERC) Consolidator Grant "Deep Learning 2.0" (grant no. 101045765). Funded by the European Union. Views and opinions expressed are however those of the author(s) only and do not necessarily reflect those of the European Union or the ERC. Neither the European Union nor the ERC can be held responsible for them. The authors additionally acknowledge funding by The Carl Zeiss Foundation through the research network "Responsive and Scalable Learning for Robots Assisting Humans" (ReScaLe) of the University of Freiburg.

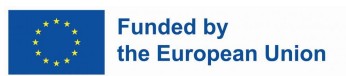

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

## A  List of Environments included in HPO-RL-Bench

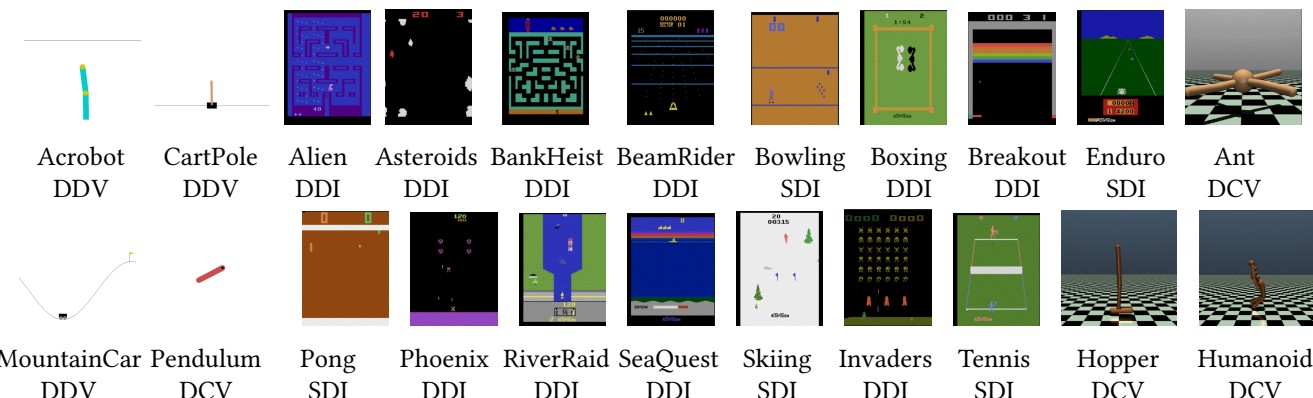

| Acrobot DDV | CartPole DDV | Alien DDI | Asteroids DDI | BankHeist DDI | BeamRider DDI | Bowling SDI | Boxing DDI | Breakout DDI | Enduro SDI | Ant DCV |

| MountainCar DDV | Pendulum DCV | Pong SDI | Phoenix DDI | RiverRaid DDI | SeaQuest DDI | Skiing SDI | Invaders DDI | Tennis SDI | Hopper DCV | Humanoid DCV |

Figure 7: Considered environments with reward, action- and state-space classification. Rewards are **D**ense or **S**parse. Action-spaces are **D**iscrete or **C**ontinuous and state-spaces are **I**mage based or **V**ector state-spaces. E.g., Pong has a sparse reward, a discrete action space and an image based state-representation. For detailed descriptions of each environment we refer to https://www.gymlibrary.dev/

## B  Data Format and Implementation Details

We store the benchmark data as a set of JSON files, separated into folders per search space and environment. Every file contains the reward curves for a hyperparameter configuration (or schedule in the case of the dynamic search space) in a given environment and search space. Specifically, the JSON file has the following fields: i) *returns_train* – the reward list returned during training, ii) *timestamps_train* – the timestamp (in seconds) associated with the training reward, iii) *timesteps_train* – the time step associated with the reward, iv) *returns_eval* – the rewards observed during evaluation and its associated measurements, v) *std_returns_eval* – the standard deviation of the evaluation reward, vi) *timestamps_eval* – the timestamp associated with the evaluation reward, vii) *timesteps_eval* – the time step associated with the evaluation reward.

We use the following naming convention for all the files in the benchmark: *%env_name%-%search_space%_**random**_%hp1%_val1%hp2%_val2_%**seed**%seedval%**eval.json***, where we apply bold fonts for fixed strings. For instance, a filename is: *BeamRider-v0_A2C_random_lr_-6_gamma_0.95_seed0_eval.json.*

We trained each configuration and seed tuple on an environment for $10^6$ steps. For every $10^4$ steps, we evaluated the agent for 10 episodes and recorded the mean and standard deviation of the obtained evaluation returns.

## C  Additional Results

### C.1  Extended Search Spaces

We have extended the search spaces from Table 1 to include architectural hyperparameters for PPO and A2C. The extended search spaces are given in Table 2.

### C.2  Hyperparameter Importance for RL Agents

Figures 15 to 17 show the average rank of best and worst hyperparameter configurations and hyperparameter importance for each search space. Figure 14 indicates a crucial insight into the relative importance of various hyperparameters for PPO and A2C. Specifically, it illustrates that the

```python
from benchmark_handler import BenchmarkHandler
from optimizers.random_search import RandomSearch

search_space ="PPO"
benchmark = BenchmarkHandler(environment="Pong-v0", search_space=search_space,
                              return_metrics=["eval_avg_returns"], seed=0)

random_search = RandomSearch(search_space=benchmark.get_search_space(search_space),
                              obj_function=benchmark.get_metrics,
                              max_budget=99)

n_iters = 100
best_conf, best_score = random_search.suggest(n_iters)
print(f"Best configuration found is {best_conf}")
print(f"Best final evaluation return is {best_score}")
```

Listing 1: Code snippet for querying HPO-RL-Bench when tuning PPO with Random Search

```python
from benchmark_handler import BenchmarkHandler

bench = BenchmarkHandler(environment="Enduro-v0", seed=0,
                          search_space="PPO", set="static")
budget = 50

# querying static configuration
config_to_query = {"lr": -6, "gamma": 0.8, "clip": 0.2}
queried_data = bench.get_metrics(config_to_query, budget=budget)
print(f'Return at budget {budget}: {queried_data["eval_avg_returns"][-1]}')

# querying dynamic configuration
bench.set = "dynamic"
config_to_query = {"lr": [-3, -4], "gamma": [0.98, 0.99], "clip": [0.2, 0.2]}
queried_data = bench.get_metrics(config_to_query, budget=budget)
print(f'Return at budget {budget}: {queried_data["eval_avg_returns"][-1]}')
```

Listing 2: Code snippet for querying HPO-RL-Bench 1.0

*number of layers*, a hyperparameter determining the number of hidden layers for the architecture of the RL algorithms, has a significant impact on their performance. In the A2C search space, the *number of layers* outranks *gamma* on average, suggesting that the model architecture can influence results more than some traditional hyperparameters. Likewise, in the PPO algorithm, the *number of layers* hyperparameter is ranked better than the *clipping range* on average. Furthermore, in the DQN algorithm, the *number of layers* and *number of units* hyperparameters are ranked better than the *epsilon* on average. This finding underlines the importance of not only tuning traditional hyperparameters but also carefully considering the architecture of the RL algorithms for achieving optimal performance.

## C.3 Performance Profiles for HPO-RL-Bench

We show performance profiles of the data included in HPO-RL-Bench for further analysis. We normalize the final evaluation return of each configuration as $\tau = \frac{return - min\_return}{max\_return - min\_return}$, where $min\_return$ represents the minimum final evaluation return, whereas $max\_return$ represents the maximum final evaluation return for a given environment and seed. Figures 19 and Figure 20 show the performance profiles of the five RL Algorithms in HPO-RL-Bench. For each RL Algorithm, the curves show the means and standard deviations of the fraction of configurations included in

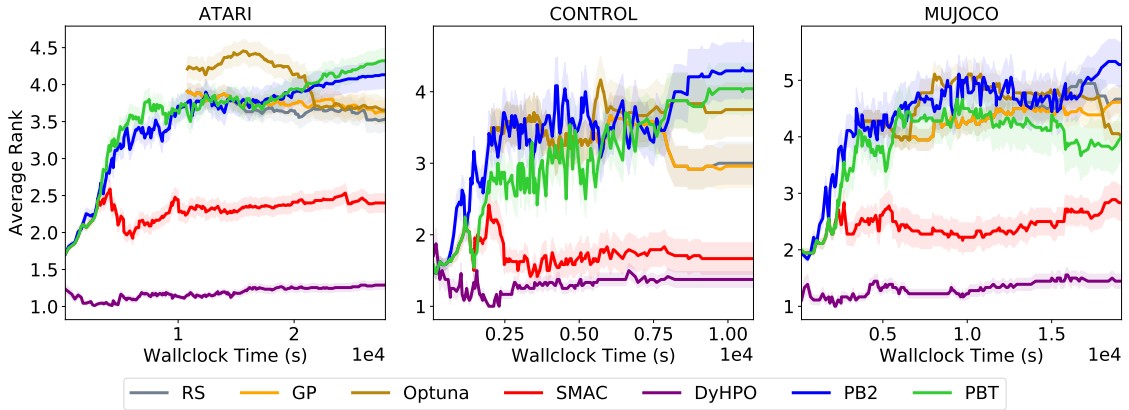

Figure 8: Average Ranks of the performance of the baselines for the A2C configuration space.

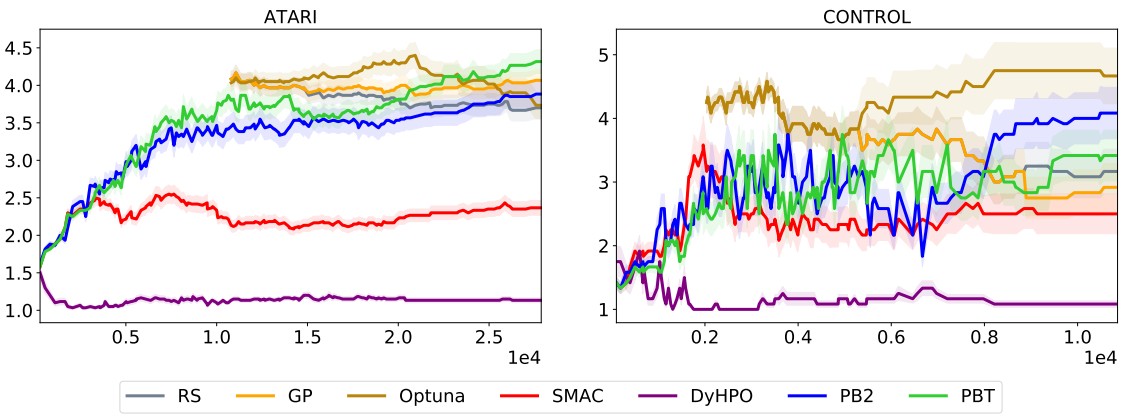

Figure 9: Average Ranks of the performance of the baselines for the DQN configuration space.

HPO-RL-Bench that have a normalization score $\tau$ at least as big as the values in the x-axis. The means and standard deviations are calculated across the seeds available in our benchmark.

This analysis allows us to get insights into the tunability of the considered agents on our chosen configuration space and also how RL algorithms compare to each other. On the MuJoCo environments, for example, all agents can be tuned to reach high performances. However, on our chosen configuration space, DDPG and TD3 are easier to tune, as they have a larger number of configurations that can achieve a high score compared to SAC, A2C, and PPO. In the classic control environments however (with the extended search space), PPO and A2C are very easy to tune as a large fraction of the configuration space results in a very good performance. SAC, TD3, and DDPG can only be tuned to achieve a maximal normalized score of around 0.8, with only very few configurations capable of achieving this score (see Figure 20)

### C.4 Rank Plots per Environment

Figures 21 and 22 show the average rank of each evaluated HPO method on the individual environments.

## D HPO-RL-Bench Reward Curves

In this section we have plotted the reward curves in HPO-RL-Bench. Figures 23- 27 show the reward curves for the PPO, A2C, DDPG, SAC, and TD3 search spaces, respectively.

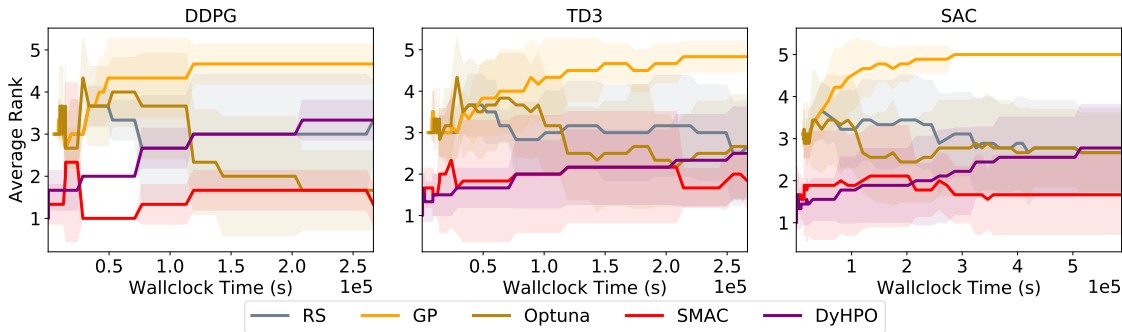

Figure 10: Average Ranks of the performance of the baselines for the DDPG, TD3, and SAC search spaces.

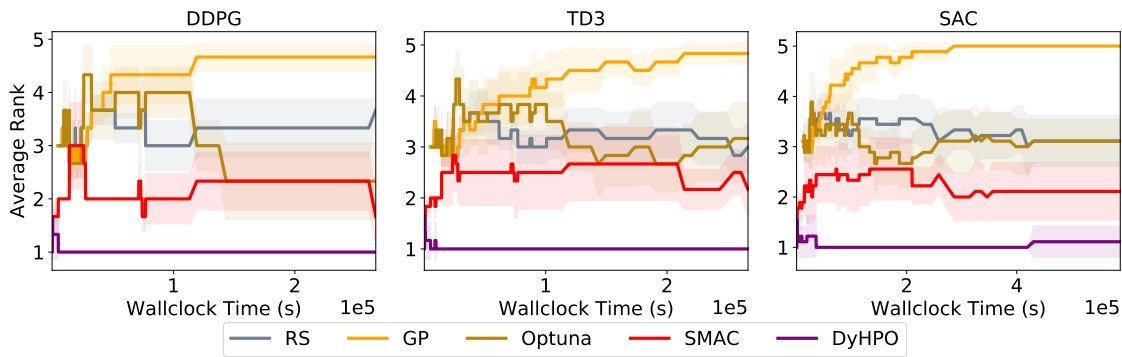

Figure 11: Average Ranks of the performance of the baselines for the DDPG, TD3, and SAC search spaces for the MuJoCo environments.

Table 2: Extended configuration spaces of PPO and A2C.

| Algo. | Hyperaram. | Values |
|-------|-----------|--------|
| PPO | lr ($\log_{10}$) | $-6, -5, -4, -3, -2, -1$ |
| | $\gamma$ | $0.8, 0.9, 0.95, 0.98, 0.99, 1.0$ |
| | clip | $0.1, 0.2, 0.3$ |
| | n_layers | $1, 2, 3$ |
| | n_units | $32, 64, 128, 256$ |
| A2C | lr ($\log_{10}$) | $-6, -5, -4, -3, -2, -1$ |
| | $\gamma$ | $0.8, 0.9, 0.95, 0.98, 0.99, 1.0$ |
| | n_layers | $1, 2, 3$ |
| | n_units | $32, 64, 128, 256$ |
| DQN | lr ($\log_{10}$) | $-6, -5, -4, -3, -2, -1$ |
| | $\gamma$ | $0.8, 0.9, 0.95, 0.98, 0.99, 1.0$ |
| | $\epsilon$ | $0.1, 0.2, 0.3$ |
| | n_layers | $1, 2, 3$ |
| | n_units | $32, 64, 128, 256$ |

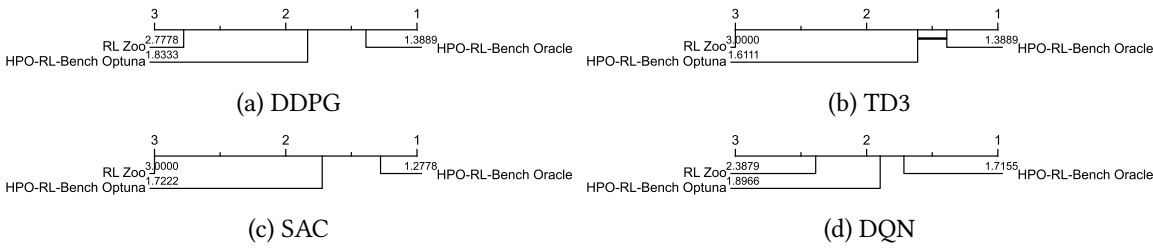

(a) DDPG

(b) TD3

(c) SAC

(d) DQN

Figure 12: Critical Difference diagram for final evaluation return of hyperparameter configurations suggested by RL-Zoo, those optimized by Optuna on the HPO-RL-Bench search space, as well as the best-performing hyperparameter configuration in HPO-RL-Bench.

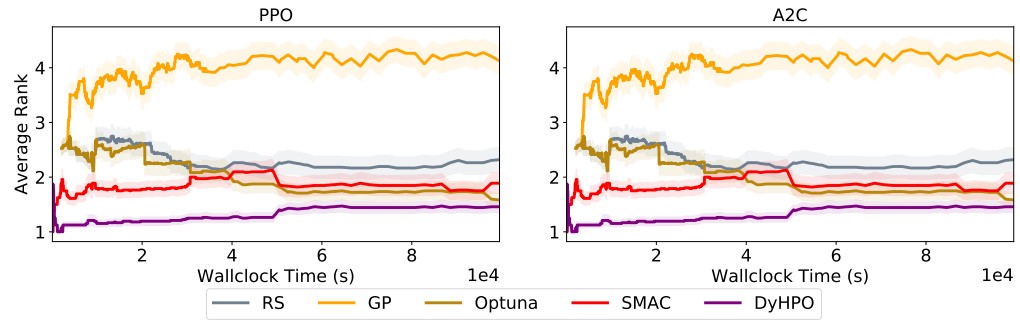

Figure 13: Average Ranks of the performance of the baselines for the extended versions of the PPO and A2C search spaces on the Classic Control environments.

## E License

We provide HPO-RL-Bench 1.0 under an MIT License. OpenAI Gym [13] and Stable-Baselines3 [50] are also offered under an MIT License.

## F Limitations and Future Work

HPO-RL-Bench provides data that allows for the evaluation of black-box HPO, grey-box HPO, as well as online HPO methods. However, we only focus on model-free RL algorithms as our search spaces. This limitation can be lifted by extending the benchmark by increasing the number of search spaces. Further, by its tabular nature, HPO-RL-Bench covers exactly the evaluated configuration

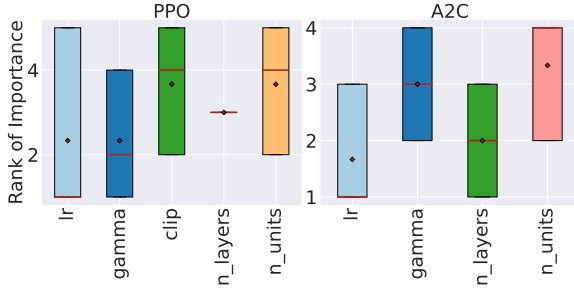

Figure 14: Hyperparameter importance for the extended versions of the PPO and A2C search spaces (red diamond=mean, red line=median, green dots=outliers, colored box=inter-quantile range(IQR) from the first quantile(Q1) to the third quantile(Q3) of the data, whiskers=extended IQR by 1.5x.).

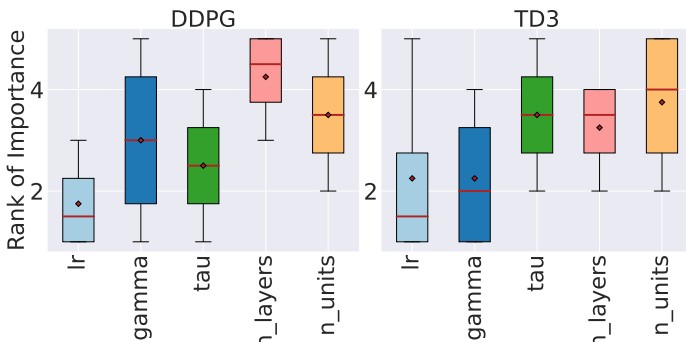

Figure 15: Hyperparameter importance per search space for DDPG and TD3(red diamond=mean, red line=median, green dots=outliers, colored box=inter-quantile range(IQR) from the first quantile(Q1) to the third quantile(Q3) of the data, whiskers=extended IQR by 1.5x.).

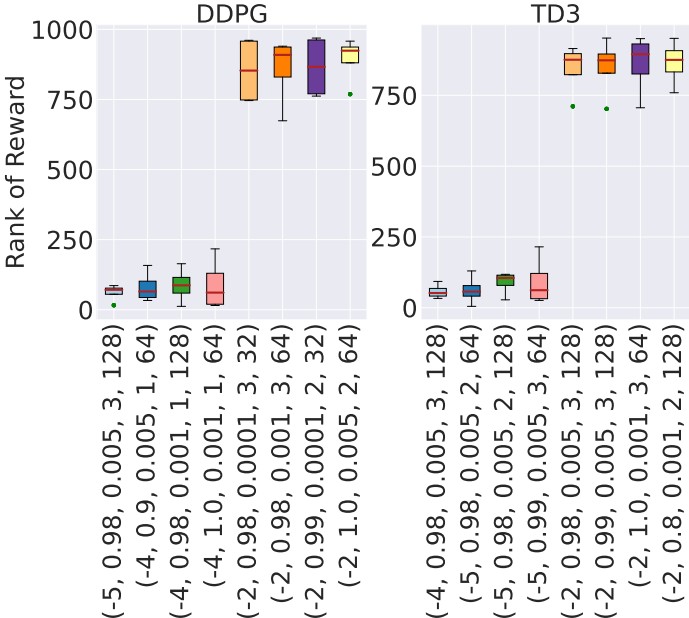

Figure 16: Average rank of the final reward across environments for TD3 and DDPG (four best and worst configurations). The illustration depicts: red diamond=mean, red line=median, green dots=outliers, colored box=inter-quantile range(IQR) from the first quantile(Q1) to the third quantile(Q3) of the data, whiskers=extended IQR by 1.5x.

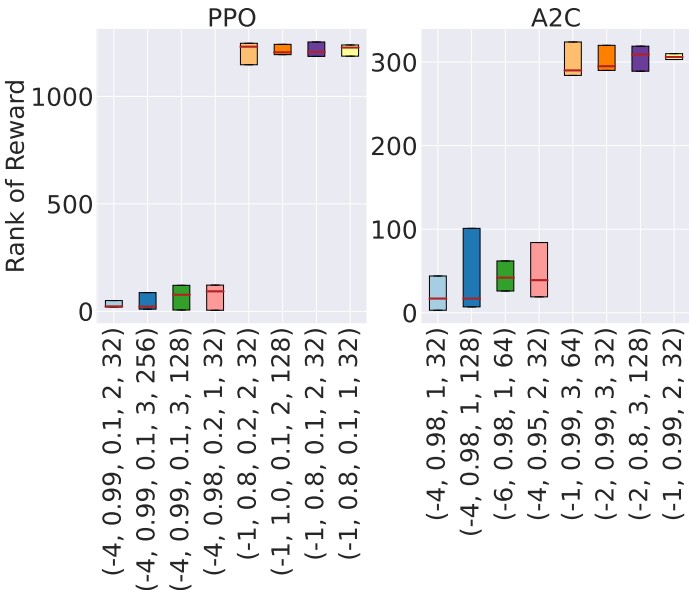

Figure 17: Average rank of the final reward across environments for the extended search spaces of PPO and A2C (four best and worst configurations). The illustration depicts: red diamond=mean, red line=median, green dots=outliers, colored box=inter-quantile range(IQR) from the first quantile(Q1) to the third quantile(Q3) of the data, whiskers=extended IQR by 1.5x.

spaces but does not allow reasoning about algorithmic behavior outside the covered space. Still, HPO-RL-Bench lays the foundation for the principled study of AutoRL and in particular HPO for RL. In future work, similar to trends in benchmarking for NAS [see, e.g., 39], we plan to use surrogate models to cover larger configuration spaces while keeping the positive aspects of a tabular benchmark.

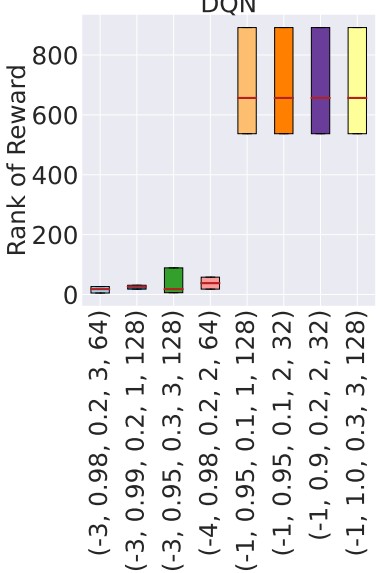

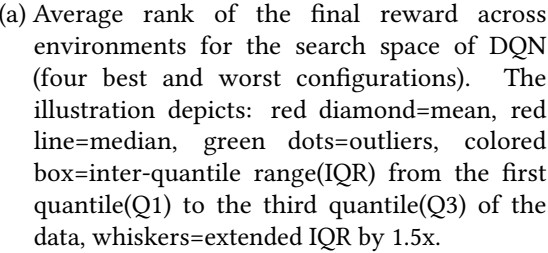

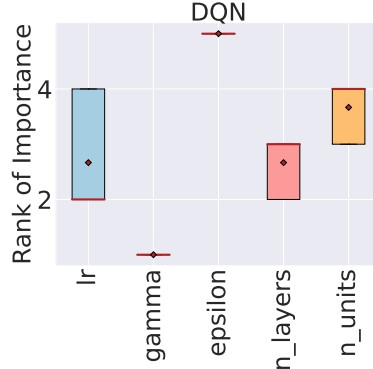

(a) Average rank of the final reward across environments for the search space of DQN (four best and worst configurations). The illustration depicts: red diamond=mean, red line=median, green dots=outliers, colored box=inter-quantile range(IQR) from the first quantile(Q1) to the third quantile(Q3) of the data, whiskers=extended IQR by 1.5x.

(b) Hyperparameter importance for DQN.

Figure 18: Hyperparameter importance analysis for the DQN search space: 18a average rank of the final reward across environments for the search space of DQN (four best and worst configurations), and 5b hyperparameter importance. The illustration depicts: red diamond=mean, red line=median, green dots=outliers, colored box=inter-quantile range(IQR) from the first quantile(Q1) to the third quantile(Q3) of the data, whiskers=extended IQR by 1.5x.

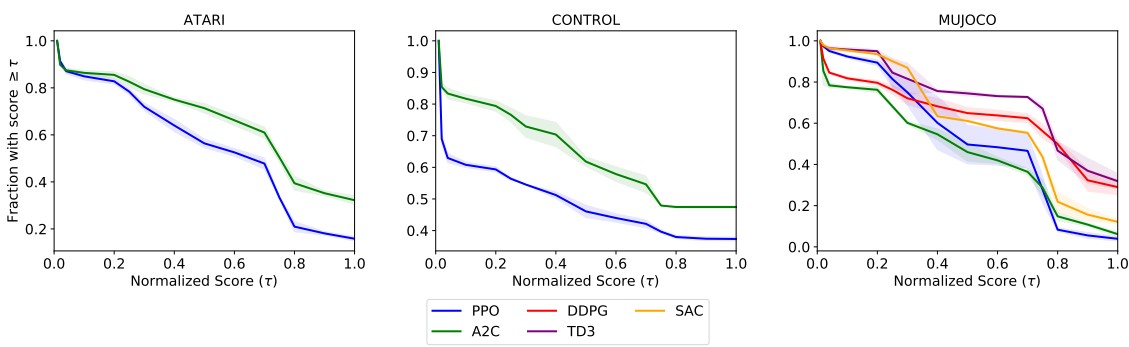

Figure 19: Performance profiles for all the configurations PPO, A2C, DDPG, TD3, and SAC on the environments in HPO-RL-Bench. The x-axis denotes values of the normalization score $\tau$, whereas the y-axis denotes the fraction of configurations that have a normalization score at least as high as the value denoted in the x-axis.

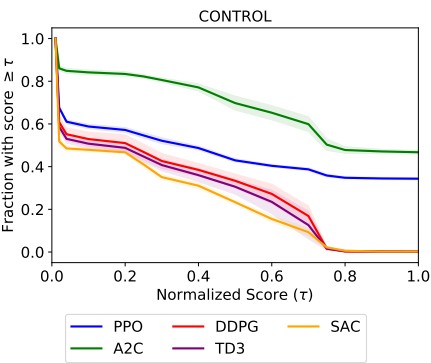

Figure 20: Performance profiles for all the configurations of the extended search spaces of PPO, A2C, DDPG, TD3, and SAC on the Classic Control environments in HPO-RL-Bench.

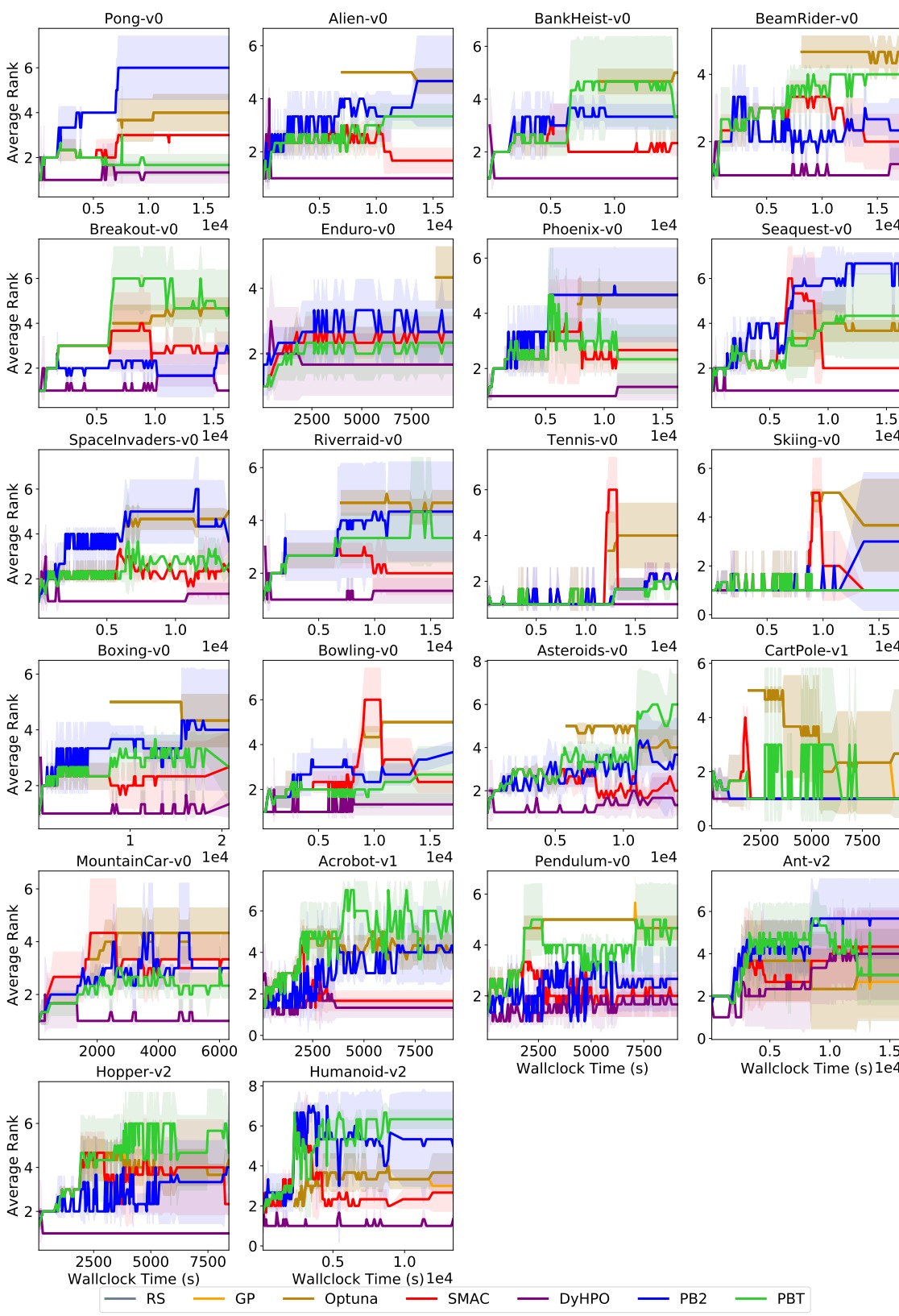

Figure 21: Average Ranks of the performance of the baselines the PPO search space per Environment.

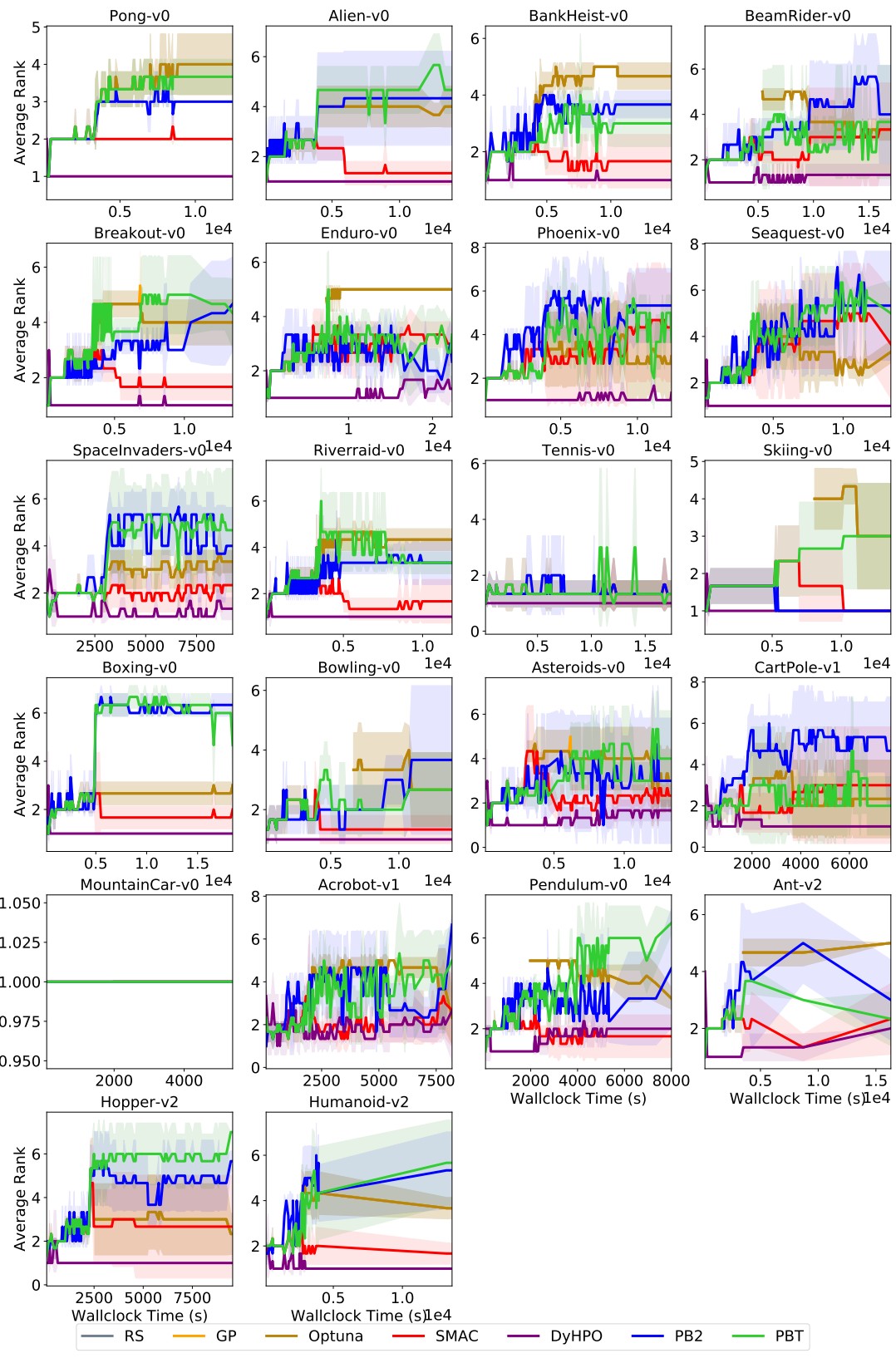

Figure 22: Average Ranks of the performance of the baselines the A2C search space per Environment.

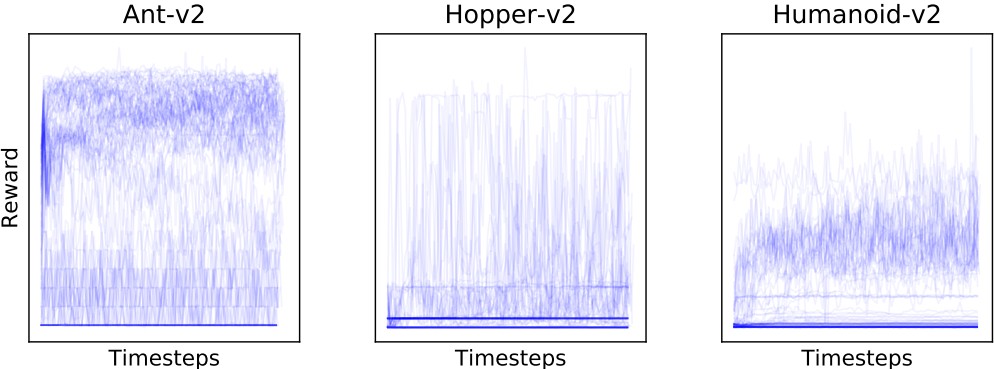

Figure 23: Reward curves of DDPG on the environments included in HPO-RL-Bench.

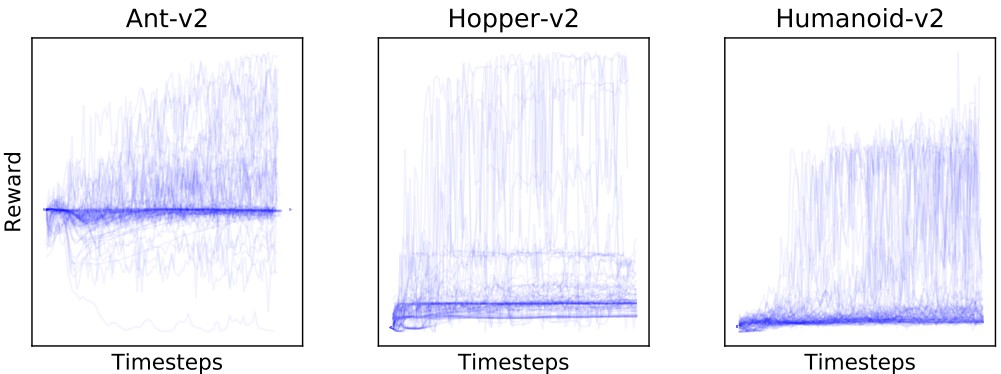

Figure 24: Reward curves of SAC on the environments included in HPO-RL-Bench.

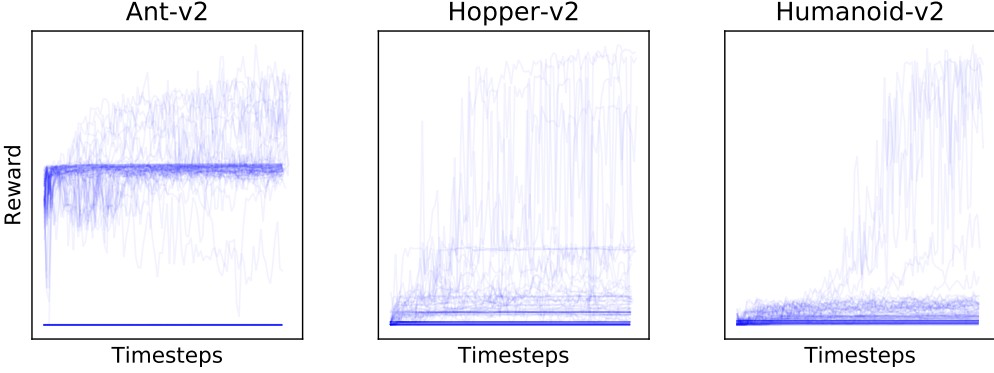

Figure 25: Reward curves of TD3 on the environments included in HPO-RL-Bench.

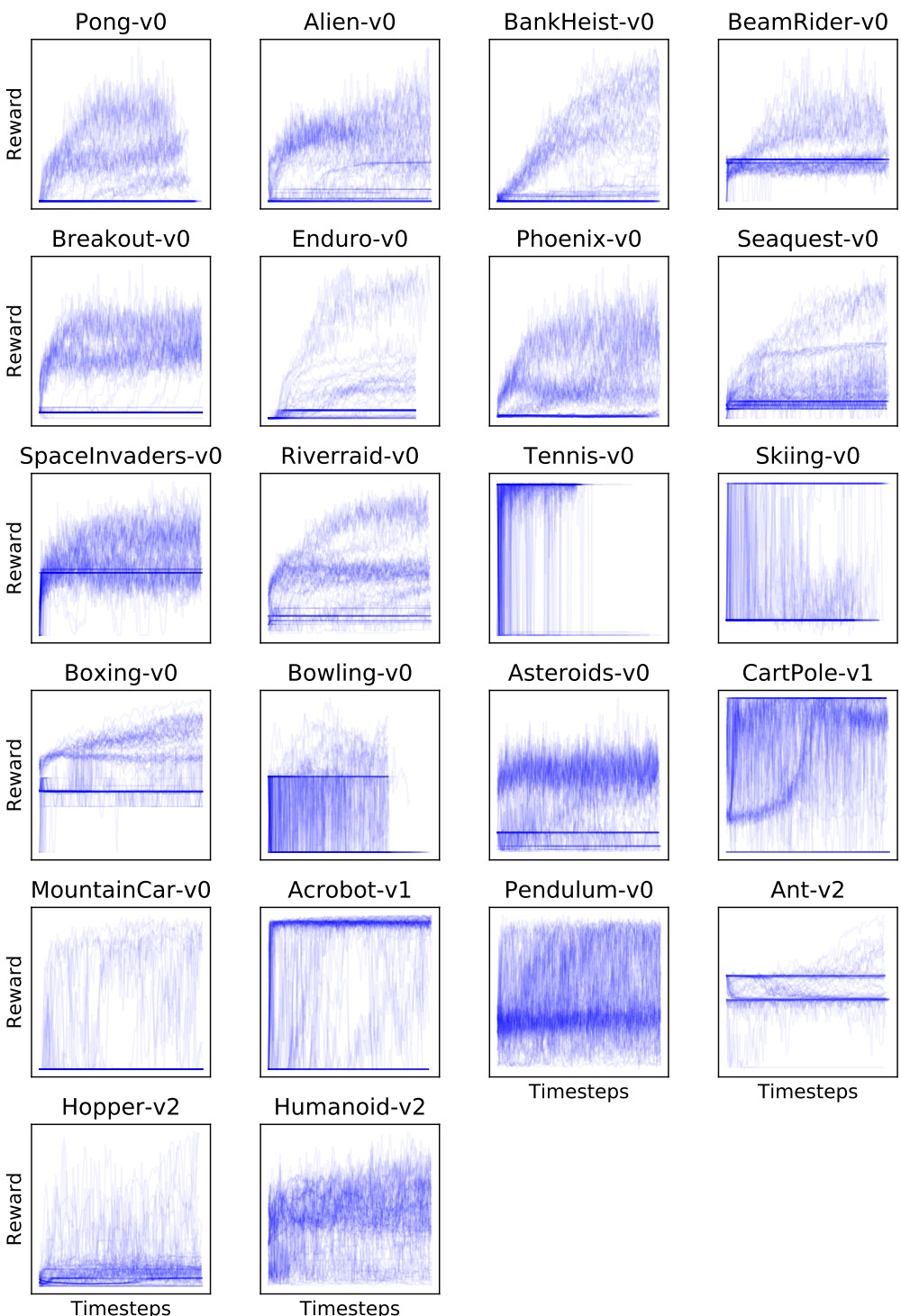

Figure 26: Reward curves of PPO on the environments included in HPO-RL-Bench.

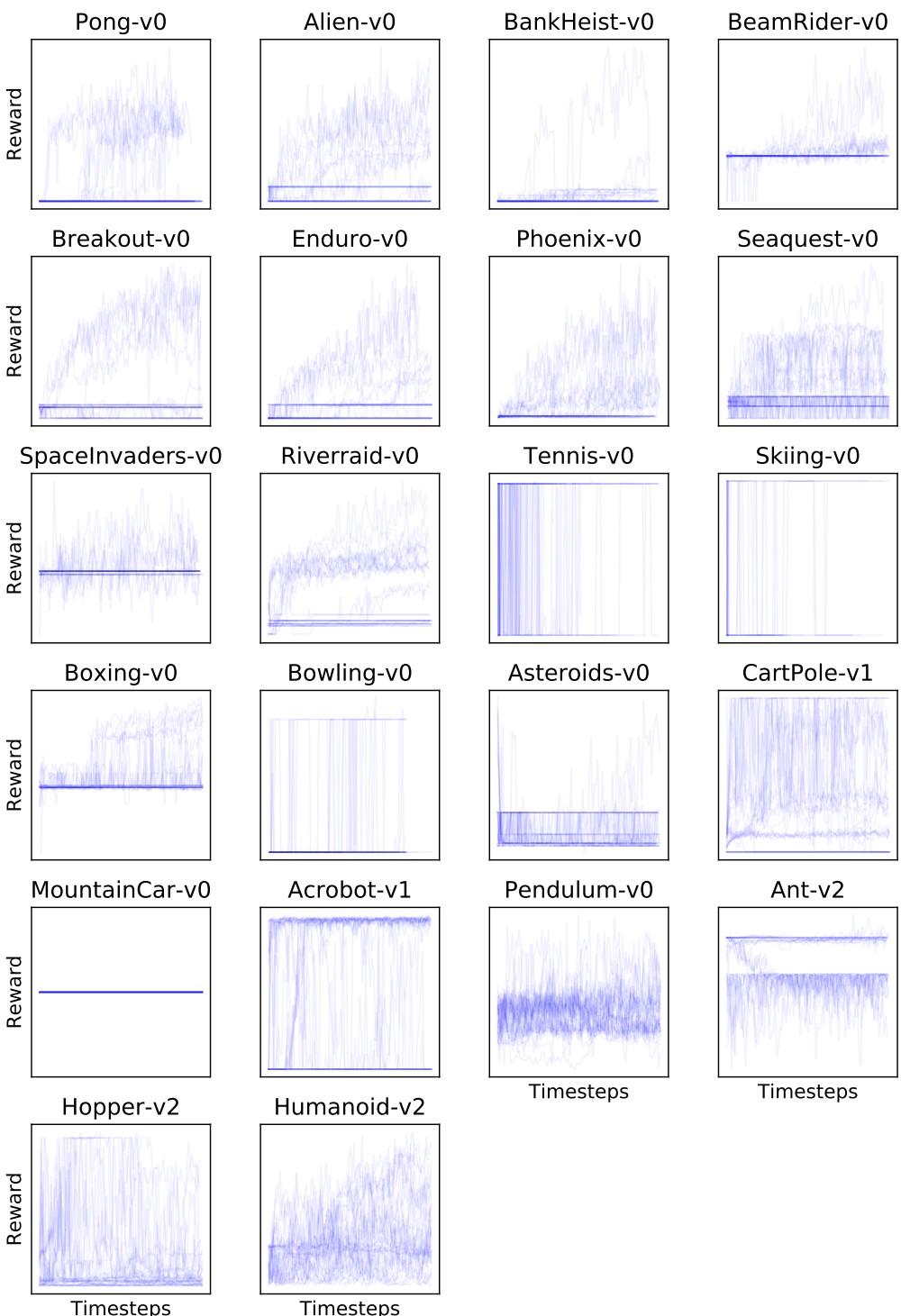

Figure 27: Reward curves of A2C on the environments included in HPO-RL-Bench.

