# OpenReview forum: "HPO-RL-Bench: A Zero-Cost Benchmark for HPO in Reinforcement Learning"
_automl.cc/AutoML/2024/ABCD_Track — AutoML 2024 (ABCD Track)_

### Official Review · Reviewer_PkzV · 2024-03-27

**Potential Impact On The Field Of Automl Rating:** 4
**Technical Quality And Correctness Rating:** 4
**Clarity:** The paper is well written.
**Clarity Rating:** 4
**Actions Required To Increase Overall Recommendation:** N/A

**Summary Of Contributions:**

This paper introduces a new benchmark for hyperparameter optimization (HPO) techniques in reinforcement learning (RL) research. The benchmark includes pre-computed reward curve evaluations of hyperparameter configurations for six RL algorithms across 22 environments. In addition, it offers plug-and-play tuning of the hyperparameters of new RL algorithms. Consequently, this benchmark enables zero-cost experiments for assessing new HPO methods in the RL space.

**Overall Review:**

Positive aspects:
- This paper is well organized, with clearly stated motivation and objective.
- The comprehensive benchmark contains multiple aspects that should be appreciated by researchers in the HPO / RL space, including but not limited to: inclusion of reward curves, tuning of hyperparameters for new RL algorithms, dynamic configuration space, etc.

Negative aspects:
N/A

**Potential Impact On The Field Of Automl:**

Yes, this paper would likely be cited for its importance in the AutoML field, especially for those researchers working on HPO in the RL field. As mentioned in the introduction, the benchmark can potentially benefit the following researchers the most: (1) HPO researcher evaluating new HPO methods in RL problem; (2) RL researcher evaluating RL algorithms against benchmark algorithms.

**Reproducibility:**

An API for the benchmark is provided with the submission. I do not see reproducibility issue with this submission.

**Review Confidence:**

4

**Review Rating:**

9

**Review Summary:**

I believe this benchmark is of significance to many areas in the AutoML field and it is employable in a fairly straightforward manner. In addition, it is a comprehensive benchmark capable of doing multiple tasks important for future research in these areas. Given these, I recommend accept for the paper.

**Technical Quality And Correctness:**

The quality of the paper is solid.

First, the benchmark provides a valuable contribution by offering a tool for studying HPO in the RL settings, which was lacking in the literature.

Second, the inclusion of reward curves for each hyperparameter configuration, problem, and algorithm is much appreciated. Such data offers valuable insights for researchers investigating early stopping policies in the RL space.

Third, the concept of a dynamic configuration space is both novel and practically important. This adds flexibility for HPO while making it applicable to real-world scenarios.

Overall, I think this is a well-thought benchmark that has significance across multiple research domains.

---

### Official Review · Reviewer_RQak · 2024-03-31

**Potential Impact On The Field Of Automl Rating:** 3
**Technical Quality And Correctness Rating:** 4
**Clarity Rating:** 3

**Summary Of Contributions:**

The study provides a large-scale benchmark of HPO for RL. Studies compare PPO, DDPG, A2C, SAC, TD3, and DQN on Atari, Mujoco, and Control environments and claim these benchmarks can directly be imported for zero-cost optimization, training of RL, and benchmarking. The study claims this is novel due to the understudy of Auto HPO in the field of RL. Benchmarks could be evaluated by other researchers at zero cost as well as provide a baseline to compare against. Library/API could help researchers tune their algorithms with already provided HPO methods.

**Actions Required To Increase Overall Recommendation:**

- Highlight the risk of the paper and provide an in-depth analysis of how this study is still useful for tasks outside of the ones that are used by the study.
- Provide legends for the box plots

**Clarity:**

The paper is easy to follow and largely free from errors. The problem is well-defined and supported by relevant research based on existing literature for HPO optimization in RL. The tables are clear and easy to follow. The paper appropriately cites existing literature when necessary. Appropriate examples and configuration parameters are provided along with an in-depth description of RL techniques when necessary.

**Overall Review:**

Given the recent growth of RL, tunning RL algorithms is a major challenge. There does not exist a study that provides precomputed reward curves for the OpenAI gym, Atari, Classic Control, and MuJoCo use cases. The study is carefully conducted and in-depth details are mentioned along with hardware requirements and experimental setup. However, these benchmarks are data-specific due to the fundamental nature of HPO. In the real world and other academic domains where data of such type is not available, these benchmarks could be irrelevant. However, these benchmarks could be used for studies based on the same datasets that the authors used. Finally, authors should provide legends for all the box plots to make it easy for the reader to compare and contrast.

[Update 1]- Authors incorporated my feedback, leading to a change in my rating

**Potential Impact On The Field Of Automl:**

HPO for RL is an important and one of the most growing fields due to the advent of recent innovations in Large Language Models and robotics. Tunning RL parameters are under-explored areas and this benchmark could help researchers to compare and contrast various learning algorithms to train. The pre-computed rewards curves could be referred to by other researchers as baselines and provides low entry to barriers for tunning new algorithms with existing HPO methods. This could help reduce time for experiments and boost research in this area.

**Review Confidence:**

3

**Review Rating:**

8

**Review Summary:**

HPO for RL is an important area and the study is meticulously conducted on 22 tasks from OpenAI gym, Atari, Classic Control, and MuJoCo. The authors provide benchmarks on these tasks that could be used off the shelf by other researchers looking to train the RL model to enable zero-cost optimization and low barriers to entry. It's important to note that HPO is data-specific and the behavior of training could completely change from data to data making the benchmark irrelevant to use cases outside the ones that the authors's analysis is based on. However, it does provide a framework to help other researchers conduct RL training and HPO. Further, Authors must update the charts with appropriate legends

[Update 1]- Authors incorporated my feedback, leading to a change in my rating

**Technical Quality And Correctness:**

The study is carefully conducted on multiple datasets with several random seeds. Baselines and reward curves are provided with confidence intervals for statistical significance. Technical details are elaborate when necessary. The reward curves are conducted on existing use cases, however it's inappropriate to generalize them to other problems. Such benchmarks are generally data-specific and interpretation might not hold true on other datasets.

---

### Official Review · Reviewer_RxeB · 2024-04-01

**Potential Impact On The Field Of Automl Rating:** 4
**Technical Quality And Correctness Rating:** 4
**Clarity Rating:** 4

**Summary Of Contributions:**

This paper introduces a new benchmark for HPO in reinforcement learning. Importantly, this is a tabular benchmark which enables zero-cost evaluation for future researchers. Aditionally, it analyses the results and provides insights on the importance of hyperparameters in RL.

**Actions Required To Increase Overall Recommendation:**

I only have very minor criticisms mentioned above. While addressing these will improve the paper, I am not sure if the impact is wide enough within the entire AutoML community to warrant the highest possible rating. Great paper nonetheless.

**Clarity:**

The work is clearly presented, with only minor adjustments that could be made to improve the clarity of figures.
The text describes the necessary details, and includes full info on search spaces, RL algorithms and HPO methods.

**Overall Review:**

RL is known to be unstable and exhibit high variance in results. It is therefore important for RL research to include robust HPO. This new benchmark can accelerate progress towards robust RL by enabling zero-cost evaluations on common tasks and algorithms. Due to the cost of training RL algorithms, quick evaluations can be powerful in this setting. The impact is therefore high.

The proposed benchmark is large and can be usedful both for HPO researchers and for RL researchers. The API for this benchmark is easily accessible and seems very intuitive.

Additionally, this paper includes useful insights on e.g. how hyperparameters tend to be environment-specific and must be individually tuned.

Minor notes.
1. In Related Work, the paragraph headings end with a colon followed by a full stop. Just remove one.
2. The text in Fig. 6 is small, especially the numbers.

**Potential Impact On The Field Of Automl:**

RL is known to be unstable and exhibit high variance in results. It is therefore important for RL research to include robust HPO. This new benchmark can accelerate progress towards robust RL by enabling zero-cost evaluations on common tasks and algorithms. Due to the cost of training RL algorithms, quick evaluations can be powerful in this setting. The impact is therefore high.

**Review Confidence:**

4

**Review Rating:**

9

**Review Summary:**

RL struggles with robustness and HPO is often overlooked. This benchmarks can help advance this area of research and lower the barrier of entry. Overall this is a high quality paper with a high potential impact and I recommend acceptance.

**Technical Quality And Correctness:**

The benchmark is generated by training algorithms implemented in common libraries. The usefulness of the proposed hyperparameter search spaces is then verified, and found to be highly competitive. The claims are well evidenced and confirmed by statistical tests.

---

### Meta-Review · Area_Chair_fGy5 · 2024-04-22

**Paper Recommendation:** Accept
**Confidence:** 4

**Metareview:**

The paper introduces a new tabular benchmark for HPO within Reinforcement Learning (RL). Reviewers are all aligned on the thorough experimental evaluation, the paper's high quality and contribution to addressing a critical gap in the RL literature, hence unanimously recommend acceptance.

---

### Decision · Program_Chairs · 2024-04-29

**Decision:**

Accept

**Comment:**

Thank you for submitting your paper. We are happy to tell you that we accept your paper to the main track. See you in Paris.